# Multiomic Approach for Bioprospection: Investigation of Toxins and Peptides of Brazilian Sea Anemone *Bunodosoma caissarum*

**DOI:** 10.3390/md21030197

**Published:** 2023-03-22

**Authors:** Maria Eduarda Mazzi Esquinca, Claudia Neves Correa, Gabriel Marques de Barros, Horácio Montenegro, Leandro Mantovani de Castro

**Affiliations:** 1Department of Biological and Environmental Sciences, Bioscience Institute, Sao Paulo State University (UNESP), Sao Vicente 11330-900, SP, Brazil; 2Biodiversity of Coastal Environments Postgraduate Program, Department of Biological and Environmental Sciences, Bioscience Institute, Sao Paulo State University (UNESP), Sao Vicente 11330-900, SP, Brazil; 3NGS Soluções Genômicas, Piracicaba 13416-030, SP, Brazil

**Keywords:** sea anemone, omics, toxins, intracellular peptides, neuropeptides

## Abstract

Sea anemones are sessile invertebrates of the phylum Cnidaria and their survival and evolutive success are highly related to the ability to produce and quickly inoculate venom, with the presence of potent toxins. In this study, a multi-omics approach was applied to characterize the protein composition of the tentacles and mucus of *Bunodosoma caissarum*, a species of sea anemone from the Brazilian coast. The tentacles transcriptome resulted in 23,444 annotated genes, of which 1% showed similarity with toxins or proteins related to toxin activity. In the proteome analysis, 430 polypeptides were consistently identified: 316 of them were more abundant in the tentacles while 114 were enriched in the mucus. Tentacle proteins were mostly enzymes, followed by DNA- and RNA-associated proteins, while in the mucus most proteins were toxins. In addition, peptidomics allowed the identification of large and small fragments of mature toxins, neuropeptides, and intracellular peptides. In conclusion, integrated omics identified previously unknown or uncharacterized genes in addition to 23 toxin-like proteins of therapeutic potential, improving the understanding of tentacle and mucus composition of sea anemones.

## 1. Introduction

Sea anemones are sessile invertebrates of the phylum Cnidaria that have inhabited the earth for over 700 million years. Their survival and evolutive success were highly related to the ability to produce and quickly inoculate venom. Cnidarians have a specialized cell responsible for venom production: the cnidocyte. This cell carries an organelle called cnida that penetrates the target to release the venom as soon as a movement is detected. Venom production is a powerful mechanism of defense, intra- and interspecific competition, and predative behavior [1,2]. A sea anemone’s venom is enriched with biomolecules, such as polypeptides, proteins, polyamines, and salts, [3], and it represents an important weapon directly associated with their ecological success. Some of these biomolecules are potent toxins able to interact with sodium and potassium ion channels [4,5] by blocking or reducing their sensibility, leading to diverse effects on the prey, including the inability to move fast.

Biomolecules produced by marine organisms have been studied for several decades, and important drugs have been obtained to treat a variety of human diseases [6,7,8,9,10]. For example, Toxin ShK, a sequence of 34 amino acids, is a potent potassium channel inhibitor isolated from *Stichodactyla helianthus* [11]. Dalazatide, a compound derived from this toxin, is undergoing a clinical trial phase 1b due to its ability to inactivate Kv1.3 channels in human T lymphocytes, whose expression is upregulated in autoimmune diseases such as psoriasis [12,13]. Another sea anemone toxin, BgK, acts on channels Kv1 and 3 in addition to Shaker-related Kv1 channels [14] and belongs to the same family of ShK.

*Bunodosoma caissarum* (*B. caissarum*) is a Brazilian sea anemone species that populates the intertidal zone of the southern coast. In the 1990s, *B. caissarum*’s extract was thoroughly studied by Malpezzi and Freitas research group, which published several findings related to its antimitotic, antitumor, cytotoxic, hemolytic, and neurotoxic activities [15,16,17,18], and described the first three toxins of this species [19]. However, access to different databases shows an absence of genome annotation for the genus of *B. caissarum*, and the number of deposited genomes for the phylum Cnidaria is still relatively low. This information is scarce when compared with the number of species estimated in the phylum—above 13,000 [20,21].

Recently, the advances in next-generation sequencing techniques, improvements in mass spectrometry instrumentation for protein sequencing, and also bioinformatic tools led to an increase in the number of large-scale molecular studies in anthozoans, using transcriptomic (RNA-seq) and/or proteomic approaches. *Exaiptasia diaphana, Nematostella vectensis, Edwardsiella lineata, Anthopleura elegantissima, Stichodactyla helianthus*, and *Stichodactyla haddoni* are some sea anemone species that have already been investigated [21,22,23,24,25,26,27]. These omics have provided a detailed overview of biomolecules, revealing a high complexity of the venom compared with results obtained in previous studies.

The understanding of the marine environment and its rich biodiversity still faces many obstacles for molecular, evolutionary, morphological, or even conservation studies. The lack of molecular information on marine fauna hinders the construction of phylogenies, the search for alternative drugs, and the analysis of the different adaptive mechanisms of these species. This study applied integrated omics approaches, including transcriptomics, proteomics, and peptidomics, to characterize the tentacles and mucus composition of *B. caissarum*.

## 2. Results

### 2.1. Tentacle Transcriptome

Sequencing data were deposited at the National Center for Biotechnology Information (Bioproject: PRJNA939171; Biosample: SAMN33456513). Transcriptome assembly yielded 186,978 transcripts from 166,364,140 paired reads. Protein prediction using TransDecoder resulted in 150,493 peptides from 75,867 complete proteins. Other metrics are represented in Table 1. Most of the genes annotated through BLASTx and BLASTp searches showed a high percentage of identity with the recently deposited draft genome of the sea anemone *Actinia tenebrosa* (*A. tenebrosa*) [28].

Sequences obtained by Trinity were searched against the Metazoa protein database. Annotation performed by Trinotate resulted in 23,444 annotated genes, categorized according to Gene Ontology terms. A total of 19,561 genes were assigned biological process classes, 19,933 were assigned cellular component classes, and 21,195 were assigned molecular function classes (Figure 1). In biological processes, the subcategories with the most genes were cellular and metabolic processes and biological regulation. In molecular function, genes related to binding, catalytic activity, and molecular transduction were more frequently found. Genes classified as subcategories of cellular components presented a homogeneous distribution. From this annotation, 1% of genes were identified as toxins (Figure 1). Initially, along with transcriptome sequences, 48 transcripts were identified with sequences similar to toxins produced by Cnidarian species, corresponding to 11 families according to the UniProtKB database, with 3 of them previously described from *B. caissarum.*

In addition, Trinity sequences obtained after the Trinotate annotation were searched against the Animal Toxin Annotation Project (Toxprot, accessed on 1 December 2021) using BLASTx local alignment tool with an e-value of 1E-04 Top BLAST hits were selected for each gene, and toxins were classified according to their families or main molecular function available on UniProtKB. A total of 2055 transcripts presented similarity with Toxprot sequences, distributed in 74 classes of toxins; nineteen of these classes with the highest number of transcripts are shown in Figure 2A. The category named ‘Others’ represents the toxin families with fewer transcripts found, such as actinoporins, sea anemone toxin 8, venom proteinase, and cysteine-type endopeptidase inhibitor (Figure 2A and Appendix A). Of these 2055 genes, only 130 transcripts showed similarity with cnidaria toxins and venom-related proteins. These transcripts represent 50 different proteins present in the UniProtKB, 95% of which were from the following sea anemones: *Nematostella vectensis*, *A. tenebrosa*, *Stichodactyla gigantea,* and *Bunodosoma caissarum*. Sea anemone toxins were grouped into three main categories: neurotoxins, mixed-function enzymes, and membrane-active. Neurotoxins are molecules able to interact with ion channels, impacting the transmission of impulses, and they include families such as potassium channel toxins, sodium channel inhibitors, and small cysteine-rich proteins from the phylum Cnidaria [29,30]. Mixed-function enzymes display important cellular roles, not necessarily related to a toxic effect. Phospholipases A2 hydrolyze phospholipids, thus directly affecting membrane configuration. These enzymes are necessary for the cell because they may act as toxins [31]. Membrane-active toxins also interact with cellular membranes, for example, porins induce cell lysis by pore formation [32,33]. Transcripts related to two types of porins were identified: actinoporins and jellyfish toxins. The proteins with more genes associated were Nematocist express protein 6 (30 genes), toxin Bcs III 15.09 (11 genes), and *Urticina crassicornis*’ phospholipase A2 (6 genes). The unknown category gathers families whose functions are not clear or well described, such as acrohargins, sea anemone 8 toxin, EGF domain peptide, and toxins that were not characterized as any family member (Figure 2B).

### 2.2. Mucus and Tentacle Proteome

Proteomic analysis was performed on tentacles and mucus protein extracts, with a total of eight independent runs in the mass spectrometer, with six samples of the fraction above 10 KDa (three from tentacles and three from mucus), and two samples of the fraction below 10 KDa (two mixtures from all tentacles and all mucus samples) as shown in Appendix A.

A total of 1580 proteins were found in tentacles or mucus samples considering an FDR < 1% and using a database with protein sequences generated by Transdecoder from the transcriptome experiment data of *B. caissarum* tentacles (Appendix A). Of these proteins, 836 were found only in tentacles, 210 only in mucus, and 534 were common in both (Figure 3A). Moreover, 66.2% of them are known and approximately one-third (34.8%) are unknown or uncharacterized (Figure 3A). Known proteins were classified by family, class, or main molecular function according to UniProtKB and then grouped into six main categories: structural, enzymes, DNA- and RNA-associated proteins, transport and signal, protein metabolism, and toxin. Some proteins were associated with more than one category. Nonrelated proteins with these functions were grouped into others. Enzymes were the most abundant category, with 377 proteins classified as follows: transferases (45), hydrolases (44), kinases (30), metalloendopeptidases (28), and peptidases (25) (Figure 3A).

To quantitatively assess differences in protein profile composition between mucus and tentacle, the proteins found in at least two of three mucus or tentacle samples (a total of 430) were selected and compared (Appendix A). In the tentacle, 316 proteins found were present exclusively or in greater quantities than in the mucus, with almost 64% of them matching a known protein (Figure 3B). In the mucus, 114 proteins were present exclusively or in greater quantities than in the tentacle, with almost 55% of them known (Figure 3C). The known proteins were classified considering their main biological function or family according to the UniProtKB database. Known tentacle proteins were mostly enzymes (30.7%), followed by DNA- and RNA-related, while in mucus, known proteins were mostly toxins (29%) (Figure 3B,C). This quantitative analysis allows us to compare 24 transcripts with similarities to toxins, as shown in Figure 4.

### 2.3. Mucus and Tentacle Peptidome

#### Non-Reduced and Non-Alkylated Samples

A total of 681 peptides from tentacle and mucus non-reduced and non-alkylated samples were identified and quantified through the Transdecoder database; they were related to 84 transcripts and 72 described proteins (Appendix A). Peptides mass distribution showed a higher abundance of fragments between 1100 and 2100 Da (Figure 5A). Most peptides found were fragments from proteins, such as actin, tubulin, myosin, histones, microtubule-associated, and hemicentin. Quantitatively, most peptide fragments were more concentrated in the tentacle fraction than in the mucus. However, it is possible to notice that for the same precursor protein, some peptides were more present in one fraction than in the other (Appendix A). Furthermore, a transcript called TRINITY_DN1239, with a higher abundance of peptides (251 sequences, with a range of 8 at 40 amino acid residues) was observed in all peptidome and proteome samples. This transcript does not match any protein of the SWISS-PROT database. Meanwhile, in the BLASTp and BLASTx, this transcript is identified as antigen-like. Another transcript with no match to any annotated gene or protein, called TRINITY_DN942, was the second precursor with more peptides, totalizing 59 fragments. In addition, peptides related to transcripts with ShK and factor 5/8 C domains, which are common among toxins, were observed, but most of the sequences identified corresponded to the initial C-terminal region (Appendix A).

### 2.4. Peptidome with Reduced and Alkylated Cysteine Residues

In the initial peptidomic experiments from tentacle and mucus, a low identification of cysteine residues was observed, showing a technical limitation in the mass spectrometry fragmentation of peptides with the presence of disulfide bridges between cysteine residues. In order to solve this problem, two additional samples were prepared (one containing a mixture of tentacle samples and the other with mucus samples), with the following additional steps: cysteine reduction using dithiothreitol (DTT), alkylation with iodoacetamide (IAA) and without trypsin treatment. This procedure causes a modification in cysteine residues, preventing the formation of disulfide bonds and consequently facilitating their detection in mass spectrometry. Unlike proteomics, trypsin was not used here to preserve naturally-generated peptides.

After this procedure, 950 different peptides were identified: 459 were exclusive to the tentacle pool and 449 were exclusive to the mucus pool (Appendix A), with 51 common peptides between the samples (Appendix A). Furthermore, the tentacle contained 105 peptides with cysteine residues, while the mucus contained 108 of them. Most of these sequences corresponded to toxin-like sequence transcripts. Regarding the size of the detected peptides, there was an increase in the number of peptides in almost all mass ranges, with a range between 500 and 4300 Da (Figure 5A). Table 2 and Table 3 show the peptide sequences that were detected with greater intensity. In mucus samples, three of the highest identified peptides matched *B. caissarum*’s toxin, U-AITX-Bcs2a, which was not observed in high expression in the tentacle. However, the tentacle showed the presence of another toxin, U-AITX-Aeq6a, also known as Acrorhagin II, which has an LD50 of 80 μg/kg in crabs [34].

### 2.5. Analysis of Cleavage Site in the Peptidome

In order to understand aspects related to proteolytic processing, the amino acid sequence of the precursor proteins and peptides generated were analyzed. Amino acids which preceded (upstream) and succeeded (downstream) peptide sequences, corresponding to the cleavage site (P1) were quantified in relation to the total number of peptides found in the procedures performed. In non-reduced and non-alkylated peptidome samples, the most frequent amino acid residue at the upstream position adjacent to the peptide sequences generated was tyrosine, followed by glutamic acid, phenylalanine, and methionine, showing a homogeneous distribution among these amino acids. Meanwhile, at the downstream position of cleavage the predominant amino acid residue present was serine, followed by leucine, and alanine. In reduced and alkylated peptidome samples, the most frequent amino acid in the upstream position was tyrosine, followed by methionine, arginine, phenylalanine, and glycine, while in the downstream position, they were serine, alanine, and glycine. Furthermore, many fragments in the alkylated and reduced samples were from the C-terminal region of the precursor proteins showing only the cleavage site corresponding to the upstream region adjacent to the N-terminal of the generated peptides (Figure 5B,C).

## 3. Discussion

One of the most significant findings of this study was an increase in the repertoire of transcripts, proteins, and peptides identified in the tentacles and mucus of the sea anemone *B. caissarum*, highlighting the identification of new sequences similar to toxins, neuropeptides, and also peptide fragments of resident proteins in the cytosol, nucleus, and mitochondria, through the data integration obtained from a multi-omic approach. De novo transcriptomics allows the identification of low-expressed genes without an available genome, while proteomics and peptidomics confirm the transcriptome data in addition to providing information about post-translational modifications, such as N-terminus acetylation. Moreover, as shown in Figure 6, the integrated analysis of the data set from the omic techniques used here demonstrates an enrichment in the identification of molecules because some of the transcripts were detected by proteomics, and others by peptidomics. These approaches together generate robust data on a poorly explored Brazilian sea anemone species, in which only nine toxin sequences have been characterized [6,19,35,36,37].

### 3.1. Considerations about the Mucus Collection Protocol and Omic Techniques

An important point to be mentioned here is the use of a protocol to release the content present in cnidae and its characterization through proteomics and peptidomics. Our analysis showed a marked enrichment of toxins, specific enzymes, and a few “housekeeping” proteins in the mucus samples in relation to the tentacles, allowing differentiation of proteins that could be related to components of the venom and those that are not. A similar procedure was performed in the sea anemone *Anthopleura dowii,* and enrichment with toxins and venom-related proteins was observed in the mucus. However, toxins were absent in tentacle proteomics, probably due to the discharging of cnidae after stress manipulation [38].

In the transcriptome data, many transcripts from the tentacle of *B. caissarum* showed a high percentage of identity with transcripts of closely related species, such as *A. tenebrosa* [28]. Nevertheless, few of these transcripts presented 100% identity. Moreover, when transcripts were related to toxins, the percentage of identity with sequences from other species was decreased, showing differences between them. These data reinforce the need to use specific databases for further proteomics analyses to search for new polypeptide sequences with biological activity.

Additionally, for the identification of toxins or proteins related to the venom, the data obtained from the transcriptome were compared with the sequences deposited in the UniProt Animal Toxin Annotation Project (https://www.uniprot.org/help/Toxins (accessed on 1 December 2021)). The search was performed in all the databases and was restricted to only described toxins in the phylum Cnidaria. There were discrepancies among the results of Toxprot analysis as shown in Figure 2. When the search was completed in Toxprot, 2055 transcripts were identified as related to toxins, and the most abundant class was latrotoxins (19%), followed by serine-type endopeptidase (15%) and others (11%) (Figure 2A). When the analyses were carried out on the phylum Cnidaria, only 130 transcripts showed similarity with 50 toxin sequences present in the database. Thus, the search carried out in the Toxprot database must be analyzed carefully. Some of the toxin sequences may contain common motifs found in other proteins, as observed in latrotoxins. This class of toxins, from spiders of the genus *Latrodectus*, is involved in neurotransmitter liberation and has ankyrin repeats, which are present in many other intracellular proteins [39,40]. Madio et al.’s (2017) analysis of the venom of the sea anemone *Stichodactyla haddoni* found similar results, pointing to 52 families of toxins as false positives after analysis of proteomics data, including latrotoxins, which were classified, in fact, as proteins with ankyrin repeats [27]. In our study, approximately half of cnidarian transcripts identified in the transcriptome were confirmed in proteomics and peptidomics analysis. These data demonstrate that transcriptome analysis used alone can lead to false positive results.

Proteome showed a considerable proportion of unknown or uncharacterized proteins (34%), which were highly identifiable in peptidome and proteome. The mostly identified was TRINITY_DN1239, characterized as major antigen-like due to its similarity with an *A. tenebrosa’s* protein. Considering protein roles, enzymatic activity was the main category among the 1580 proteins identified in all samples, and 316 highly expressed in the tentacle, while in mucus, toxin-like were more common among its 114 highest expressed proteins. According to Toxprot alignment with those proteins, the tentacle had approximately 1.9% of its proteins related to toxins, and mucus had 14%. The difference between these proportions corroborates mucus’s ecological importance for sea anemones’ survival and evolution [41,42].

Another aspect and the main differential applied here, compared with other integrated omic studies, was the inclusion of peptidome. Unlike proteome, where a set of peptides is generated in vitro through digestion with a specific enzyme, such as trypsin, for broad identification of protein sequences, the peptidome reflects the naturally generated peptides, adding information referent to proteolytic processing. Peptidome is restricted to molecules with less than 10 kDa, and most studies in different types of samples usually do not include the reduction and alkylation of cysteine residues in the protocol. However, cysteine residues are present in many described toxin sequences, so two protocols were performed for the peptidome to increase the number of sequenced fragments: with and without reduction and alkylation, both without trypsin. In this case, peptidomics allowed the identification of the larger and smaller fragments of the mature toxin. To exemplify, peptide fragments identified in proteomics and peptidomics for one toxin-like protein are shown in Figure 7. In this case, peptidomics allowed the identification of the larger fragment of the mature toxin, as well as smaller fragments.

### 3.2. Tentacle and Mucus Composition

The transcriptome carried out in this study aimed at obtaining a specific database, which could be used as a reference database for sequencing proteins and peptides, correlating with proteomics and peptidomics data obtained by mass spectrometry. In this context, and taking into account the diversity of genes and peptides found, the results will be discussed below in the following subsections: toxins, cysteine-rich proteins, neuropeptides, and intracellular peptides (InPeps).

#### 3.2.1. Toxins

Toxins detected in the annotated transcriptome and confirmed in the proteome, reduced and alkylated peptidome, and/or non-reduced and non-alkylated peptidome in more than one sample were selected and are listed in Table 4. Transcripts with toxin domains that presented sequences in the other omic approaches that were identified with a PFAM (Protein Families Database) search were selected and are presented in Table 5.

The most abundant toxin family identified was sea anemone type 3 (BDS) potassium channel toxin with 7 proteins codified by 9 transcripts (Table 4). This family’s members share a defensin domain (PF07936.13). According to Beress et al. (1985), toxins from this family have antihypertensive and antiviral activity. U-AITX-Bcs2a, already characterized from *B. caissarum* [6], is one of the members. Two proteins matched the described toxins from *Bunodosoma granulifera*: U-AITX-Bgr3a and U-AITX-Bgr3d, which can selectively inhibit voltage-gated potassium channels type Kv11/KCNH/ERG according to UniProt database. Matches for Kappa-AITX-Avd4m and U-AITX-Ael2c, which is an APeTx-like peptide, were also found. In the reduced and alkylated peptidome, a toxin codified by TRINITY_DN4279 with a defensin domain and no similarity to any other described toxin is a candidate for the type 3 potassium channel family (Table 5).

The Kunitz domain is a highly conserved region common in toxins, characterized by serine protease inhibition, which is observed among sea anemone peptides. Four toxins belonging to the sea anemone potassium channel type 2 subfamily were identified: π-stichotoxin-Hcr2e, U-actitoxin-Avd3n, π-actitoxin-Aeq3a-like, and kπ-actitoxin-Avd3e-like (Table 4). π-SHTX-Hcr2e (also called InhVJ) inhibits trypsin and chymotrypsin, but not Kv channels. U-AITX-Avd3n or AsKC11 is an activator of G-protein-coupled inward-rectifier potassium channels, which regulates cellular excitability in the brain and cardiac cells. It also acts in voltage-gated potassium (Kv) 1.2 and dendrotoxin receptors [43,44]. Other proteins with the Kunitz domain were identified in proteomic data, such as Carboxypeptidase inhibitor SmCl-like from *A. tenebrosa* (Table 5).

Peptides with cysteine patterns are of increasing interest because of their higher stability, giving them resistance to proteolysis. The epidermal growth factor domain is a disulfide-rich structure highly abundant among the plant and animal kingdoms. Diverse toxins from the EGF domain peptide family are deposited in the UniProt database from sea anemones *A. tenebrosa, Nematostella vectensis, Stichodactyla haddoni, Bunodosoma caissarum* and ants *Myrmecia gulosa* and *Manica rubida*. Recently, Eagles et al. (2022) described ant toxin MIITX2-Mg1a’s activity as mimetic of EGF-like hormones, engendering a long-lasting hypersensitivity in mice [45]. One transcript confirmed in the proteome matched the described Gigantoxin 4 or U-AITX-avd12a from *Anemonia viridis* and a putative toxin with a similarity to neurogenic locus notch homolog protein 1 presented EGF-domain peptides.

Proteome and peptidome also contained turripeptides, members of the conopeptide P-like family, from *Polystira albida* and *Lophiotoma olangoensis*, two marine gastropods. Cone snails, as well as sea anemones, produce a wide range of biomolecules with high selectivity, turning them into potential pharmaceuticals. Turripeptides contain a Kazal-type domain, which is common in protease inhibitors, having six cysteine residues; turripeptides are also capable of ion channel regulation. These peptides were described in zoanthids, jellyfish, and marine annelids, but not in sea anemones [46,47,48,49]. Six transcripts have the Kazal-type domain, according to PFAM. Three were classified as putative toxins; one of them, identified as Agrin, is a protein that induces aggregation of nicotinic acetylcholine receptors [50]; one with fibrilin-2-like isoform X3; and one with neurogenic locus notch homolog protein 1-like (Table 5). Two new toxins with Kazal domain were identified by their similarity to π-AITX-Avd5a and turripeptides Pal 9.2 (Table 4), which were also found in the reduced and alkylated peptidome.

Phospholipases A2 cleaves glycerophospholipids, membrane components, generating lysophospholipids and fatty acids that compose the venom of snakes, scorpions, spiders, bees, corals, and sea anemones [51,52,53,54,55,56]. One phospholipase A2 was identified, but did not share higher similarity to the phospholipase A2, already characterized from *B. caissarum*’s, phospholipase A2 Bcs-2a. These sequences shared higher similarity to *A. tenebrosa*’s basic phospholipase A2 pseudexin A chain-like.

ShKT domain was described after the *Stichodactyla helianthus*, ShK, and is characteristically present among potassium channel inhibitors in sea anemones. The Actinaria family, which presents the ShKT domain, is potassium channel toxin type 1, and in the proteome, peptides with similarity to U-actitoxin-Avd8a-like, Kappa-actitoxin-Aer3a (AETX-K) and Kappa-actitoxin-Bgr1a (BgK) were identified. The U-AITX-Avd8a toxin belongs to the sea anemone 8 toxin family, whose functions remain poorly described. AETX-K acts in Shaker and Shaw Kv channels, as well as BgK, which inhibits Kv 1.1, 1.2, and 1.3 channels largely distributed in the central nervous system [14,57]. In addition to these described toxins, six transcripts were identified and confirmed in the proteome and in the reduced and alkylated peptidome with ShK domain, all of which had a similarity to uncharacterized proteins of *A. tenebrosa*. These sequences were classified as new putative toxins since they have the ShK domain and were identified in other sea anemone species.

**Table 4 marinedrugs-21-00197-t004:** *Bunodosoma caissarum*’s new toxins identified through similarity with other described toxins in transcriptome and confirmed in proteome and/or reduced and alkylated peptidome. * These toxin names were suggested according to Oliveira et al. (2006) proposed nomenclature for sea anemone toxins [58].

Toxin	Trinity Code	Transcript (Identity)	E-Value	Toxin Family	Tentacle/Mucus	Omic Approach
* U-actitoxin-Bcs3d	TRINITY_DN72667	U-actitoxin-Bgr3d (42.3%)	4.4E−13	Sea anemone type 3 (BDS) potassium channel toxin	−/+	Proteome, RA peptidome
* U-actitoxin-Bcs3a	TRINITY_DN10015 TRINITY_DN5917 TRINITY_DN38124	U-actitoxin-Bgr3a (38.1%)	4.8E−06	Sea anemone type 3 (BDS) potassium channel toxin	+/+	Proteome, RA peptidome
* U-actitoxin-Bcs2c	TRINITY_DN42790	U-actitoxin-Ael2c (45.7%)	1.8E−09	Sea anemone type 3 (BDS) potassium channel toxin	+/+	Proteome
* Κ-actitoxin-Bcs4m	TRINITY_DN4181	Kappa-actitoxin-Avd4m (50%)	1.4E−16	Sea anemone type 3 (BDS) potassium channel toxin	+/+	Proteome, RA peptidome
U-actitoxin-Bcs2a	TRINITY_DN14686	U-actitoxin-Bcs2a (100%)	1.3E−30	Sea anemone type 3 (BDS) potassium channel toxin	+/+	Proteome, RA peptidome
Bcs Tx3	TRINITY_DN1024	type III potassium channel toxin protein, partial (96%)	1.4E−47	Sea anemone type 3 (BDS) potassium channel toxin	+/+	Proteome, RA peptidome
* Κ-actitoxin-Bcs2a	TRINITY_DN58216	Kappa-actitoxin-Ael2a (50%)	1.2E−08	Sea anemone type 3 (BDS) potassium channel toxin	−/+	RA peptidome
* ΚΠ-actitoxin-Bcs3e	TRINITY_DN13254	kappaPI-actitoxin-Avd3e-like (73.5%)	4.9E−90	Venom Kunitz type. Sea anemone type 2 potassium channel toxin.	+/+	Proteome
* Π-actitoxin-Bcs2e	TRINITY_DN217	PI-stichotoxin-Hcr2e (56.4%)	2.5E−28	Venom Kunitz type. Sea anemone type 2 potassium channel toxin.	+/−	Proteome
* U-actitoxin-Bcs3n	TRINITY_DN6572	U-actitoxin-Avd3n (74.7%)	1.6E−51	Venom Kunitz type. Sea anemone type 2 potassium channel toxin.	+/+	Proteome, RA peptidome
* Π-actitoxin-Bcs3a	TRINITY_DN10770	PI-actitoxin-Aeq3a-like (85.9%)	1.2E−59	Venom Kunitz type. Sea anemone type 2 potassium channel toxin.	−/+	Proteome
* U-actitoxin-Bcs12a	TRINITY_DN7989	U-actitoxin-Avd12a (74.4%)	9.4E−38	EGF-domain peptide	+/+	Proteome
Turripeptide Pal 9.2-like	TRINITY_DN4295	Turripeptide Pal 9.2 (38.2%)	1.2E−12	Kazal-type serine protease inhibitor domain	+/+	Proteome, RA peptidome
* Π-actitoxin-Bcs5a	TRINITY_DN713	PI-actitoxin-Avd5a (67.4%)	8E−24	Kazal-type serine protease inhibitor domain	+/+	Proteome
Basic phospholipase A2 pseudexin A chain-like	TRINITY_DN24551	Basic phospholipase A2 pseudexin A chain-like (72.8%)	7.8E−108	Phospholipase A2	−/+	Proteome
* Κ-actitoxin-Bcs1a	TRINITY_DN3227	Kappa-actitoxin-Bgr1a (89.2%)	1.4E−20	Sea anemone type 1 potassium channel toxin	−/+	Proteome
* Κ-actitoxin-Bcs3a	TRINITY_DN4127	Kappa-actitoxin-Aer3a (55.4%)	1.3E−28	Sea anemone type 1 potassium channel toxin	+/+	Proteome
* U-actitoxin-Bcs8a	TRINITY_DN1939	U-actitoxin-Avd8a-like (82.3%)	1.9E−51	Sea anemone 8 toxin	+/+	Proteome
* Δ-actitoxin-Bcs2a	TRINITY_DN24411	Delta-stichotoxin-Sgt2a (48.2%)	3.1E−15	Anemone neurotoxin	−/+	Proteome, RA peptidome
Zinc-metalloproteinase nas-13-like	TRINITY_DN24501	Zinc-metalloproteinase nas-13-like (84.9%)	3.5E−80	Astacin	−/+	Proteome
Zinc metalloproteinase nas-4-like	TRINITY_DN2272	Zinc metalloprotease nas-4-like isoform X2 (82.9%)	4E−269	Astacin	+/+	Proteome
Zinc metalloproteinase nas-15-like	TRINITY_DN1535	Zinc-metalloproteinase nas-15-like (68.2%)	1.2E−221	Astacin	+/+	Proteome
Cystatin	TRINITY_DN6968	Cystatin-like (57.2%)	7.3E−65	Cystatin domain	+/+	Proteome

RA peptidome = Reduced and alkylated peptidome; NRNA peptidome = Non-reduced and non-alkylated peptidome. Bgr: *Bunodosoma granuliferum*; Ae: *Actinia equina; Av:Anemonia viridis;* Bc: *Bunodosoma caissarum;* Hc: *Heteractis crispa;* Sg: *Stichodactyla gigantea.*

**Table 5 marinedrugs-21-00197-t005:** *Bunodosoma caissarum*’s new toxins identified through similarity with other described toxins in transcriptome and confirmed in proteome and/or reduced and alkylated peptidome.

Trinity Code	Transcript (Identity)	E-Value	Toxin Family	Tentacle/Mucus	Omic Approach
TRINITY_DN4279	-	-	Sea anemone type 3 (BDS) potassium channel toxin	+/+	Proteome, RA peptidome
TRINITY_DN14881	Carboxypeptidase inhibitor SmCl-like (64.1%)	3E−32	Venom Kunitz type. Sea anemone type 2 potassium channel toxin	−/+	Proteome
TRINITY_DN4129	Neurogenic locus notch homolog protein 1 (37.1%)	7.9E−12	EGF-domain	+/+	Proteome
TRINITY_DN3459	Agrin-like (42.5%)	4.1E−71	Kazal-type serine protease inhibitor domain	−/+	Proteome
TRINITY_DN322	Fibrilin-2-like isoform X3 (69.5%)	4.3E−70	Kazal-type serine protease inhibitor domain	+/+	Proteome
TRINITY_DN14498	Neurogenic locus notch homolog protein 1-like isoform X6 (81.8%)	2.4E−55	Kazal-type serine protease inhibitor domain	+/+	Proteome
TRINITY_DN1312	Uncharacterized protein ZK643.6-like (47.2%)	6.4E−52	ShK domain-like	+/+	Proteome
TRINITY_DN22515	Uncharacterized protein LOC116291117 (72.1%)	3.5E−75	ShK domain-like	+/+	Proteome
TRINITY_DN10788	Uncharacterized protein LOC116293550 (67.9%)	2.2E−227	ShK domain-like	+/−	Proteome
TRINITY_DN10521	Uncharacterized protein LOC116287301 (52.2%)	1.9E−48	ShK domain-like	+/+	Proteome, RA peptidome, NRNA peptidome
TRINITY_DN4363	Uncharacterized protein LOC116301037 (80.4%)	0	ShK domain-like	+/−	Proteome
TRINITY_DN19291	Uncharacterized protein LOC116291117 (61.8%)	1.3E−135	ShK domain-like	+/+	Proteome
TRINITY_DN2326	mRNA, partial (76%)	5.9E−45	Anemone neurotoxin	−/+	RA peptidome
TRINITY_DN52114	Uncharacterized protein LOC116291022 (70.8%)	6.6E−102	F5/8 C domain	−/+	Proteome
TRINITY_DN16749	Uncharacterized protein LOC116297803 isoform X6 (67.5%)	1.4E−50	F5/8 C domain	−/+	Proteome
TRINITY_DN8245	Uncharacterized protein LOC116298543 (67.6%)	5.2E−70	F5/8 C domain	+/+	Proteome
TRINITY_DN37576	Uncharacterized protein LOC116297953 (65.3%)	6.2E−44	F5/8 C domain	−/+	RA peptidome, NRNA peptidome
TRINITY_DN10809	Perlucin-like protein (65.7%)	1.2E−98	Lectin C-type domain	+/+	Proteome

RA peptidome = Reduced and alkylated peptidome; NRNA peptidome = Non-reduced and non-alkylated peptidome.

A similar candidate to Δ-stichotoxin-Sgt2a or gigantoxin II, which is a sodium channel toxin with an LD50 of 70 μg/kg in crabs, was also found in our proteomic data. This toxin is characterized as having a sea anemone neurotoxin domain. Another putative toxin with this domain (TRINITY_DN2326) was identified in the reduced and alkylated peptidome.

C-type lectins (CTLs) are commonly found in snake venoms, and, recently, bioprospection of marine organisms also showed the presence of these lectins [59]. CTLs have a variety of functions, but the higher expression of receptors in this lectin type in infection situations suggests its role in immune responses [60]. C-type lectins are important for inflammation, phagocytosis, and antigen defense, due to their antimicrobial effects [61], including against viruses such as SARS-CoV-2 [62]. CTLs may inhibit entry, attachment, or replication of the antigen material by binding with glycoprotein from the viral envelope and preventing binding and recognition with receptors [59,63,64]. C-type lectins from diverse marine species are arousing interest among scientists due to their involvement with immune responses. Through domain detection with PFAM, a C-type lectin was classified as a putative toxin, and its presence was confirmed in the proteome (Table 5 and Figure 8).

Another domain described in putative toxins is Factor V and Factor VIII, also called F 5/8C or discoidin domain, characterized by their role in the mammalian blood coagulation system. Proteins with this domain are usually responsible for cell recognition, adhesion and aggregation, and phospholipid binding, behaving as lectins according to Baumgartner et al. (1998) [65]. Madio et al. (2017) identified F 5/8C domain sequences in *Stichodactyla haddoni*’s transcriptome, although these transcripts were not confirmed in the proteome [27]. Here, three F 5/8C were identified in the proteome and one in the reduced and alkylated peptidome (Table 5 and Figure 8); it was only possible to identify them using PFAM because all of them matched uncharacterized proteins from *A. tenebrosa*.

Metalloproteases observed in omics data belong to the astacin (M12A), reprolysin (M12B), neprilysin (M13), leucine aminopeptidase (M17), and other families, and their major role is disrupting the extracellular matrix [66]. These proteins are important components of venom. From the astacin family, proteins similar to zinc metalloproteinase-nas from *Caenorhabditis elegans* were present. Astacins, considered as toxins, have diverse functions, such as processing biological peptides, regulating collagen fibers assembly and digestion, and they are critical for development and morphogenesis [67,68,69]. ShKT domain was also observed in *Hydractinia echinate* as well as in *C. elegans* [70]. Three transcripts with similarity to zinc metalloproteinase nas-13 and 15 were identified and confirmed in the proteome. Disintegrin and metalloproteinases belong to the reprolysin or M12B family and are involved in processes such as carcinogenesis, neoplastic progression, metastasis, extracellular matrix degradation, immune response evasion, and inflammation in mammals [71]. Two transcripts presented a similarity of around 40 percent with mammals but reached 80 percent identity to *A. tenebrosa*’s proteins. Peptidase M13 domain characterizes a group of metallopeptidases, represented by neprilysin, a protein involved in the inflammatory response. Two transcripts have the M13 domain, one of them matched endothelin-converting enzyme 1-like, which is a type II integral membrane protein and is important for muscle contraction and morphogenesis [66], and the other with membrane metallo-endopeptidase-like 1; both are described from *A. tenebrosa*. The peptidase M17 domain characterizes the leucine aminopeptidase family, whose major role is in the protein degradation process, releasing the N-terminal amino acid. Two transcripts contained the domain and were highly similar to putative aminopeptidases from *A. tenebrosa* (up to 90%). Other two metalloprotease domains were identified: M1 (aminopeptidase family) with one transcript and M2—peptidyl-dipeptidase A, which cleaves dipeptides from the C-terminal—with one. A serine peptidase domain was also identified in transcripts confirmed in the proteome, the subtilase domain. Two proteins with this domain matched *A. tenebrosa*’s proteins, and one of them was also identified in the reduced and alkylated peptidome (Appendix A). Except for the astacin domain-containing proteins, all the other peptidase domains containing proteins were described as venom proteins (they are described as venom components and not as toxins).

#### 3.2.2. Cysteine Rich Proteins

Cysteine-rich proteins found in proteomic data may be classified according to their domain: cysteine-rich secretory protein family (CAP) and scavenger receptor cysteine-rich domain (SRCR). The first family is distributed all over the animal kingdom, remarkably, in snake venoms. Four cysteine-rich secretory family proteins were identified in the proteome, two of them matched Golgi-associated plant pathogenesis-related protein 1-like, a C-type lectin, also observed in *Chrysaora fuscescens*’ venom [72]. Those proteins presented higher enrichment in tentacle samples.

The scavenger receptor family is a superfamily of transmembrane cell glycoproteins related to the innate immune response, characterized by their ability to pattern recognition of diverse types of ligands. Neubauer et al. (2016) evaluated the SRCR domain proteins of six cnidaria species and observed their high importance for symbiosis regulation and immune modulation in the phylum [73]. In proteome samples, 13 proteins were identified with the scavenger receptor cysteine-rich domain and 5 were identified in the reduced and alkylated peptidome (TRINITY_DN126, TRINITY_DN19873, TRINITY_DN21663, TRINITY_DN499, TRINITY_DN9220), only 2 of them in common with proteome. Five of these proteins matched the “deleted in malignant brain tumors 1 protein-like” of *A. tenebrosa*, which was named after the “deleted in malignant brain tumors” (DMBT) from *Homo sapiens*, a protein with 8 SRCR domain repeats. As other studies suggest, the innate immune gene families are expanding in invertebrates, and as observed by Neubauer et al. (2016), with anthozoan, it is not different [73]. Our results corroborate this hypothesis. The same study also relates the presence of proteins with diverse combinations of domains with SRCR, and in the reduced and alkylated peptidome, peptides of a transcript (TRINITY_DN499) with lectin C-type, EGF-like, and SRCR domains were identified.

#### 3.2.3. Neuropeptides

Antho-RFamide neuropeptides occur in neurosecretory cells; these neuropeptides are related to photoreceptive organs in cnidarians and are classified as FMRF-amide-like peptides, or FLPs. This group of peptides controls several functions in cnidarians, such as reproduction, larval movement, metamorphosis, and muscle contraction, among others [74,75,76]. Katsukura et al. (2003) describe that RFamide neuropeptides inhibit the metamorphosis of *Hydractinia chinate*, while LWamide induce the process [77,78].

After cleavage, some neuropeptides undergo various post-translational modifications, especially at the N-terminal and/or C-terminal residues. A frequently observed modification is amidation at the C-terminus of neuropeptides, protecting them from degradation by carboxypeptidases. Pyroglutamic acid is a modification present at the N-terminus of several neuropeptides, which protects the peptide chain from enzymatic degradation by aminopeptidases. The LWamide and Antho-RFamide neuropeptides described in anemones have these modifications. As these are post-translational modifications, they are identified in mass spectrometry experiments [79,80].

Our results demonstrated the presence of Antho-RFamide neuropeptides, both in the transcriptome and in the tentacle peptidome. In the sequencing provided by the Transdecoder, it was possible to observe the presence of multiple copies of these sequences described in the literature. In the peptide extracts analyzed by mass spectrometry, 18 distinct fragments of Antho-RFamide were detected (Figure 9A). For neuropeptide LWamide, only the presence of the transcript was detected. (Figure 9B).

#### 3.2.4. Intracellular Peptides (InPeps)

Previous studies conducted in rodent tissues [81], zebrafish [82], human cell lines [83], fungi [84], and plants [85], revealed the presence of a very large set of stable and bioactive intracellular peptides [86]. Some of these peptide fragments of intracellular proteins found altered in different biological models have shown a variety of biological and pharmacological activities, such as regulating signal transduction [87], regulating glucose uptake [88], altering intracellular calcium levels [89], potentiating interferon-gamma activity [90], and inducing cell death in various tumor cell lines [91,92].

Many intracellular peptides identified in this study were from histones. In the literature, the antimicrobial activity of histone fragments, both complete sequences and N-terminal [93] or C-terminal fragments [94], have been demonstrated in fish and invertebrates [95]. There was also a fragment of histone H2B type 1-H intracellular peptide (PepH) showing a protective effect against cell death in neuro2A cells [96]. In this context, histone fragments are one of the examples that show the importance of applying extraction methodologies that prioritize the fraction below 10 kDa. The exciting and largely recognized therapeutic potential of intracellular peptides [87] suggests that pharmacological characterization of intracellular peptides identified here should still be conducted.

## 4. Materials and Methods

### 4.1. Collection and Maintenance of Specimens

The specimens of *B. caissarum* were collected during the low tide period on Itaquitanduva beach, located in the State Park “Xixová Japuí”, in São Vicente County, on the south coast of São Paulo State (−23,9991794, −46,3915857). The two collection events were conducted with an interval of 6 months and, together, totalized 15 specimens of *B. caissarum*. The sea anemones were manually collected off the rocky shore and inserted on a recipient with seawater, protease inhibitor, and aerator. The acclimatization period lasted 7 days with a photoperiod of 12 h, salinity of 33 ppm in artificial seawater, and 25 °C temperature. The specimens were fed two or three times a week with small fragments of squid. For mucus obtention, the specimens were inserted in hypertonic medium (salinity: 40 ppm) and physically stimulated with a spatula around the oral disc in three cycles of 3 min of stimulating activity and 7 min of rest. Three samples with the content of three sea anemones’ mucus each were obtained and posteriorly prepared as described in proteome protocols [38].

### 4.2. Transcriptomic

#### 4.2.1. RNA Extraction, mRNA Library Synthesis and Illumina Sequencing

For the transcriptomic approach, one specimen was inserted in ice for 5 min and the tentacles were removed with surgical microscissors and laboratory glass. After the extraction, the sample was immediately stored in liquid nitrogen for shipping to NGS Soluções Genomicas. After adding Rnase-free Dnase I in order to eliminate genomic DNA, RNA extraction was performed following the protocol based on the Rneasy Lipid Tissue Kit. RNA integrity was evaluated with Bioanalyzer. The samples presenting an optic density ratio of 260/280 nm higher than 1,8 and RNA integrity higher than 7 were selected for the following experiments. The cDNA library was synthesized with ~1 ug of total RNA, following the recommendations of the RNA TrueSeq manufacturer. The resultant cDNA libraries were sequenced using an Illumina HiSeq 2500 platform. The Real-Time Analysis program (Illumina) performed the base calling of sequencing images, converting them into fastq sequences. Before assembly, raw reads were evaluated using FastQC.

#### 4.2.2. Transcriptome Assembly, Annotation and Functional Enrichment

The resultant sequencing reads were submitted to de novo assembly using Trinity version 2.11.0, a pipeline for transcriptome assembly. Standard assembly parameters were used with a minimum length of 50 bp. Trinity filtered bases of low quality, removed Illumina adapters with Trimmomatic version 0.36, counted k-mers present in sequencing reads with Jellyfish version 2.3.0, and performed digital normalization to reduce data volume, without affecting the assembly’s accuracy.

### 4.3. Proteomic and Peptidomic

#### Protein and Peptide Fractions Preparation

Protein and peptide fractions of tentacle and mucus from *B. caissarum* were prepared as previously described with some modifications [97]. The tentacles collected were resuspended in 10 mL of deionized water at 80 °C each, homogenized, and kept at the same temperature for 20 min. After this incubation, the samples were cooled at 4 °C on ice for 40 min and then acidified with HCl to a final concentration of 10 mM. The cooled samples were sonicated 3 times with 20 pulses of 1s at 4 Hz and the homogenates centrifuged at 15,000× *g* for 40 min at 4 °C. After this point, the supernatants were filtered through a Millipore membrane that allows the passage of molecules with a molecular weight of less than 10 Kda (Amicon Ultra, Millipore, Burlington, MA, USA).

The eluates not retained by the membrane (peptide fractions) were loaded onto C18 pre-columns (OASIS—Waters, UK), washed with deionized water, and eluted within 20 and 30% acetonitrile containing 0.15% trifluoroacetic acid (TFA). Sample volumes were concentrated to 5 μL in a vacuum centrifuge and stored at −80 °C.

The samples retained by the membrane (protein fraction, to 10 Kda) were processed for proteomic analysis. A solution of 8 M Urea was added to the incubation sample to a final concentration of 4 M, followed by the addition of dithiothreitol (DTT) to a final concentration of 5 mM. The mixture was incubated at 65 °C for 60 min. Iodoacetamide (IAA) was then added to a final concentration of 15 mM, and the samples were incubated for 60 min at room temperature in the dark. To quench the excess of IAA, DTT was added to a final concentration of 10 mM; proteins were digested with 1:50 trypsin:sample (porcine; Promega, Madison, WI, USA) overnight at 37 °C, acidified with formic acid, and desalted.

### 4.4. NanoLC and Mass Spectrometry

The peptide mixture was suspended in 0.1% formic acid and analyzed as follows. An UltiMate 3000 Basic Automated System (Thermo Fisher^®^, Waltham, MA, USA) was set up and connected online with a Fusion Lumos Orbitrap mass spectrometer (Thermo Fisher^®^) at the mass spectrometry facility RPT02H/Carlos Chagas Institute—Fiocruz, Paraná. The peptide mixture was chromatographically separated on a column (15 cm in length with an internal diameter of 75 μm), packed in-house with ReproSil-Pur C18-AQ 3 μm resin (Dr. Maisch GmbH HPLC) with a flow rate of 250 nL/min of 5% to 38% ACN in 0.1% formic acid on a 120 min gradient. The Fusion Lumos Orbitrap was placed in data-dependent acquisition (DDA) mode to automatically turn between full-scan MS and MS/MS acquisition with 60 s dynamic exclusion. Survey scans (300–1500 *m*/*z*) were acquired in the Orbitrap system with a resolution of 120,000 at *m*/*z* 200. The most intense ions captured in a 2 s cycle time were chosen, excluding those which were unassigned or had a 1+ charge state. The selected ions were then isolated in sequence and fragmented using HCD (higher-energy collisional dissociation) with normalized collision energy of 30%. The fragment ions were analyzed with a resolution of 50,000 at 200 *m*/*z*. The general mass spectrometric conditions were as follows: 2.3 kV spray voltage, no sheath or auxiliary gas flow, heated capillary temperature of 175 °C, predictive automatic gain control (AGC) enabled, and an S-lens RF level of 30%. Mass spectrometer scan functions and nLC solvent gradients were regulated using the Xcalibur 4.1 data system (Thermo Fisher^®^).

### 4.5. Protein and Peptide Identification

The raw data files (.raw) generated by the mass spectrometer were searched in the Transdecoder database built from the *B. caissarum* transcriptome, using the PEAKS Studio software (version 8.5; Bioinformatics Solution, Waterloo, ON, Canada) [98,99]. The research parameters used were as follows: no enzymatic specificity; precursor mass tolerance adjusted to ±30 ppm and fragmentation ion mass (tolerance ±0.5 Da); oxidized methionine (+15.99 Da) and acetylation (+42.01 Da) were defined as variable modifications. Carbamidomethylation (+57.02) was also added as a variable modification for alkylated and reduced samples. The identified peptides were then sorted by their mean local confidence to select the best spectra for annotation and filtered by FDR ≤1%.

### 4.6. Quantitative Data Analysis

Enriched proteins of tentacle and mucus were identified according to their sample area. Proteins that presented sample area as zero were replaced by a minimum value of 1E−02 because PEAKS show detected proteins with low signal as zero and when the protein was not present in the sample, it showed as “-”. This symbol was replaced by zero and zero by 1E−02, followed by standard normalization. The mean value of each protein for tentacle and mucus samples was calculated, and proteins that were first replaced by 1E−02, were replaced with the mean value. RStudio ggplot package and Python matplotlib were used to plot data.

## 5. Conclusions

In summary, we report the first integrated omics analysis to characterize the protein composition of tentacles and mucus for the sea anemone species *B. caissarum*. The proteomics and peptidomics data obtained using the sequences generated from the tentacle transcriptome as a database proved to be a robust method in the identification of similar toxins but not completely identical to the sequences already characterized in the phylum of cnidarians, belonging to the following families: sea anemone type 3 (BDS) potassium channel toxin Venom Kunitz type, sea anemone type 2 potassium channel toxin, EGF-domain peptide, Kazal-type serine protease inhibitor domain, phospholipase A2, sea anemone type 1 potassium channel toxin, sea anemone 8 toxin, astacin and cystatin domain, among others. In addition, toxins were more present in the mucus than in the tentacle, showing that the stimulation protocol helps to distinguish which proteins are more related to the venom and which are more structural. Finally, the peptidome also revealed the presence of many fragments of intracellular proteins, neuropeptides, and toxin fragments of different sizes, showing proteolytic processing. This approach demonstrates the complexity of the venom of this species of a sea anemone.

## Figures and Tables

**Figure 1 marinedrugs-21-00197-f001:**
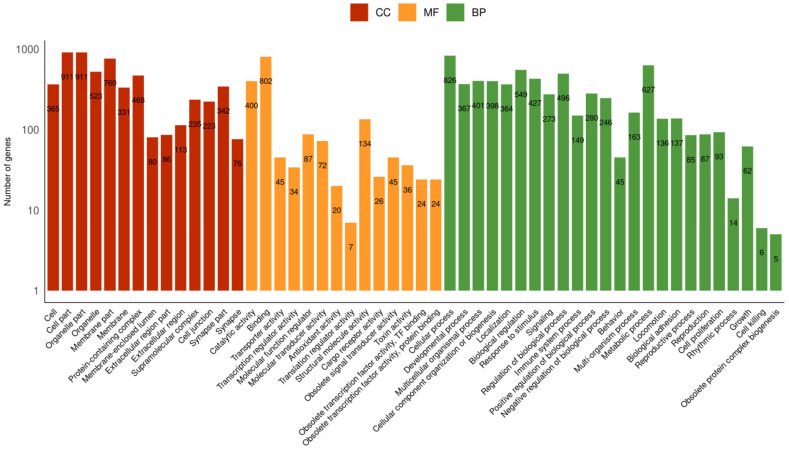
Number of annotated genes in subcategories according to Gene Ontology terms generated by Web Gene Ontology Annotation Plot (WEGO). The subcategories can be classified as cellular component (CC), molecular function (MF), and biological process (BP).

**Figure 2 marinedrugs-21-00197-f002:**
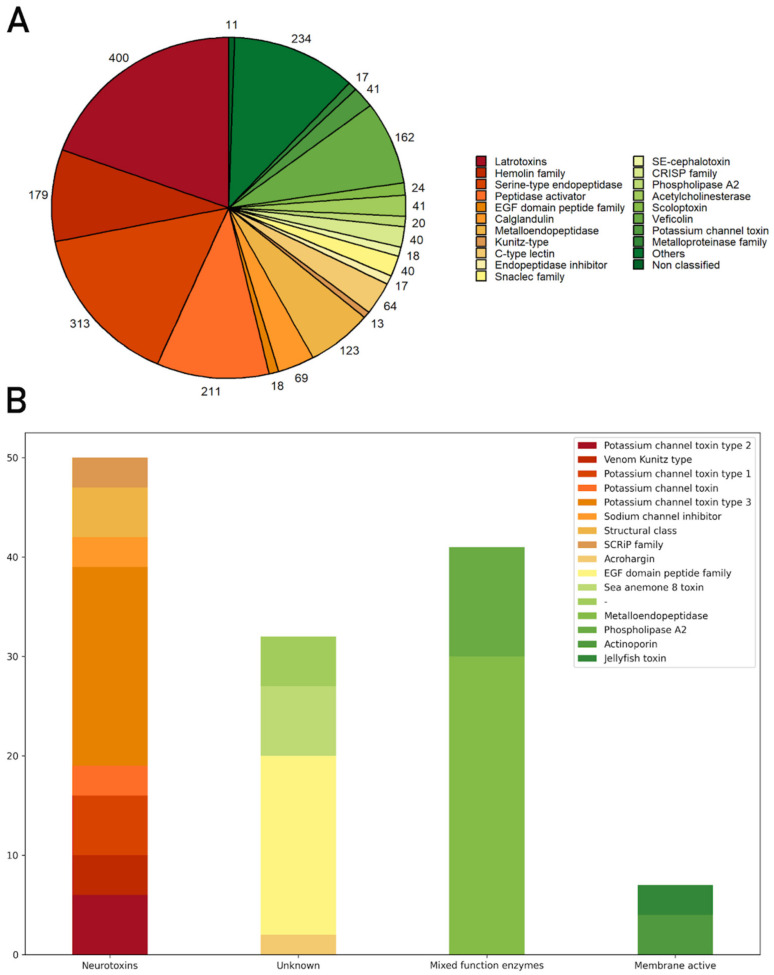
Resultant classes of toxins found in the transcriptome sequences related to the Toxprot database (**A**). Cnidaria toxin genes identified in BLASTp search of transcripts against the Toxprot database (https://www.uniprot.org/help/Toxins (accessed on 1 December 2021)) (**B**). In the graph, transcripts with similarity to toxins were grouped into four categories: neurotoxins, mixed-function enzymes, membrane-active, and unknown.

**Figure 3 marinedrugs-21-00197-f003:**
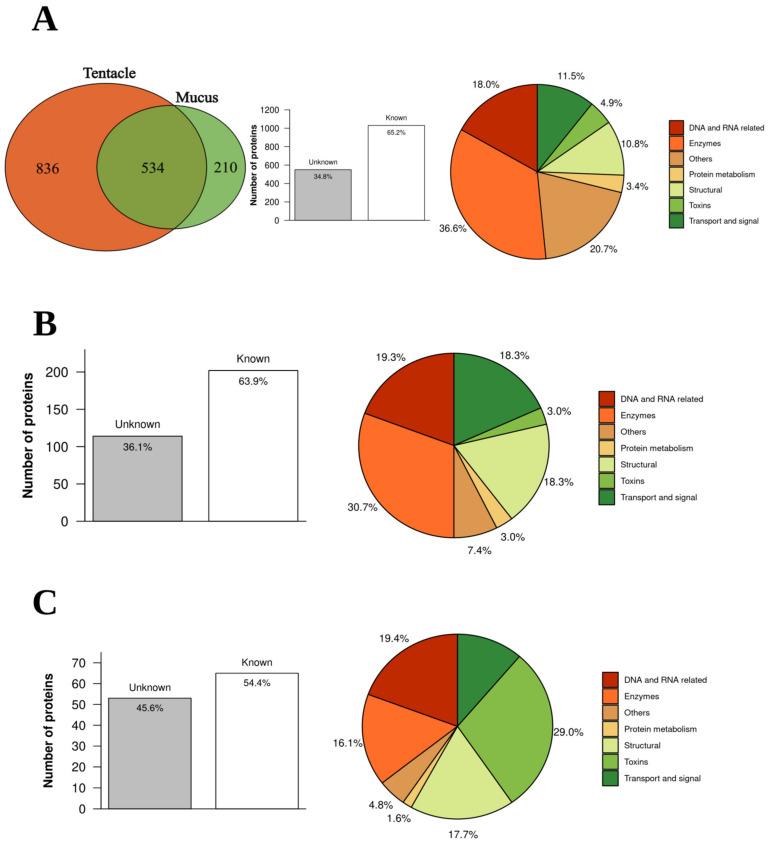
General profile of tentacle and mucus proteomics from *B. caissarum*. In (**A**) Venn diagram of proteins found in tentacle and mucus samples, percentage of known and unknown proteins, and protein classification into the following categories: structural, enzymes, DNA- and RNA-associated proteins, transport and signal, protein metabolism, and others. In (**B**) profile of enriched proteins in the tentacle. In (**C**) profile of enriched proteins in the mucus.

**Figure 4 marinedrugs-21-00197-f004:**
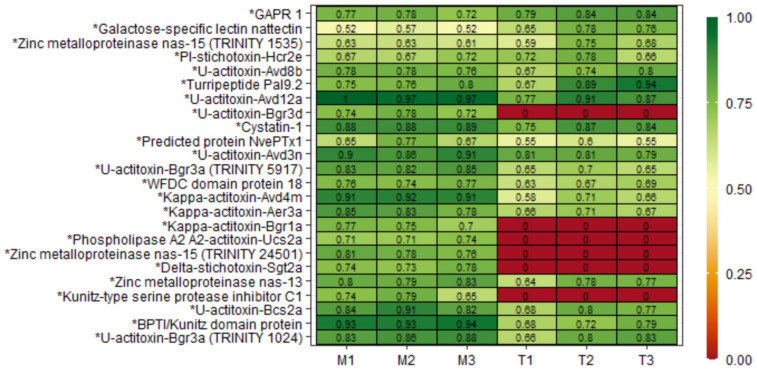
Enriched toxins were identified in tentacle and mucus samples from *B. caissarum*. Selected proteins were found in at least two out of three samples. Zinc metalloproteinase nas-15 and U-actitoxin-Bgr3a presented similarities with more than one transcript. * Putative toxins identified through similarity with other described toxins in transcriptome and confirmed in proteome.

**Figure 5 marinedrugs-21-00197-f005:**
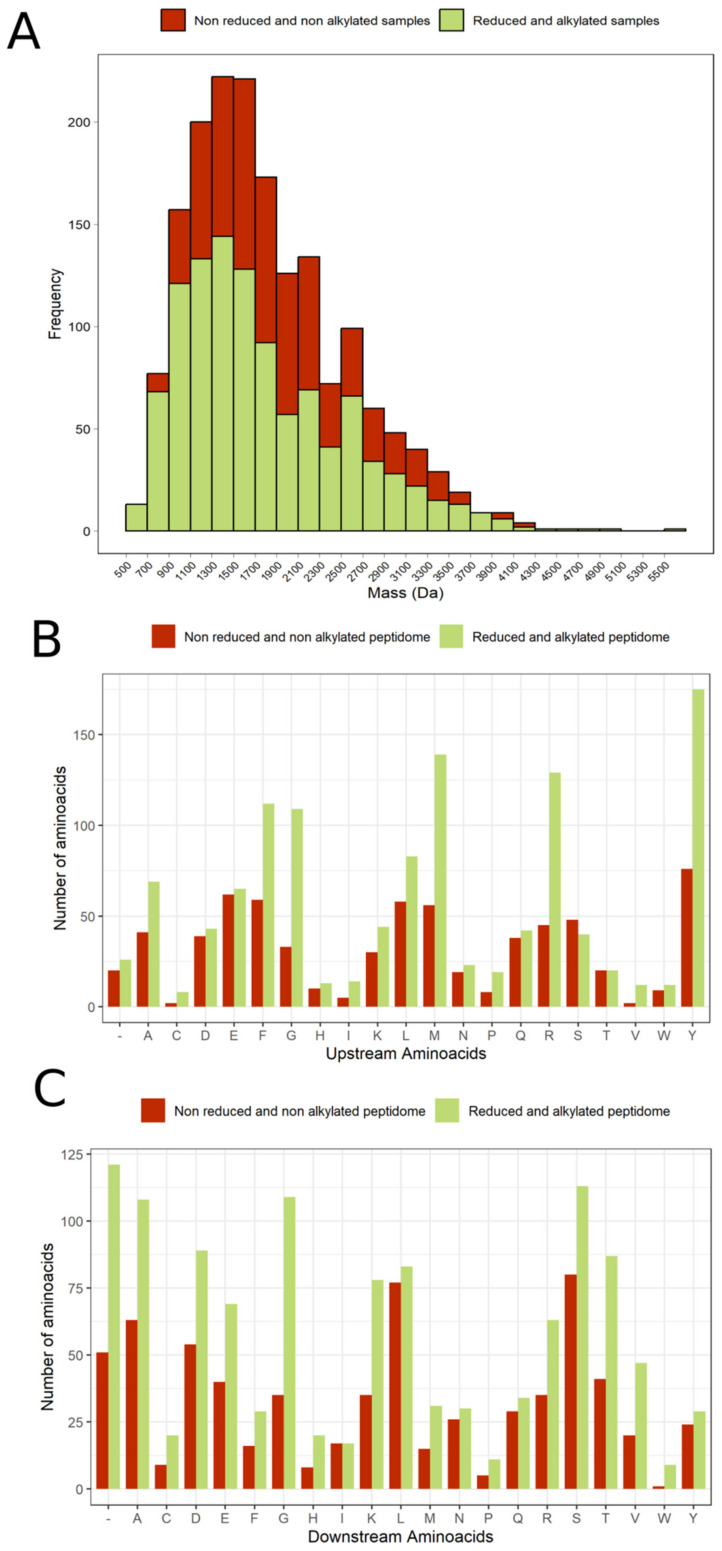
(**A**) Distribution of peptides obtained from non-reduced and non-alkylated samples (red) and reduced and alkylated samples (green). (**B**) Upstream amino acids. (**C**) Downstream amino acids.

**Figure 6 marinedrugs-21-00197-f006:**
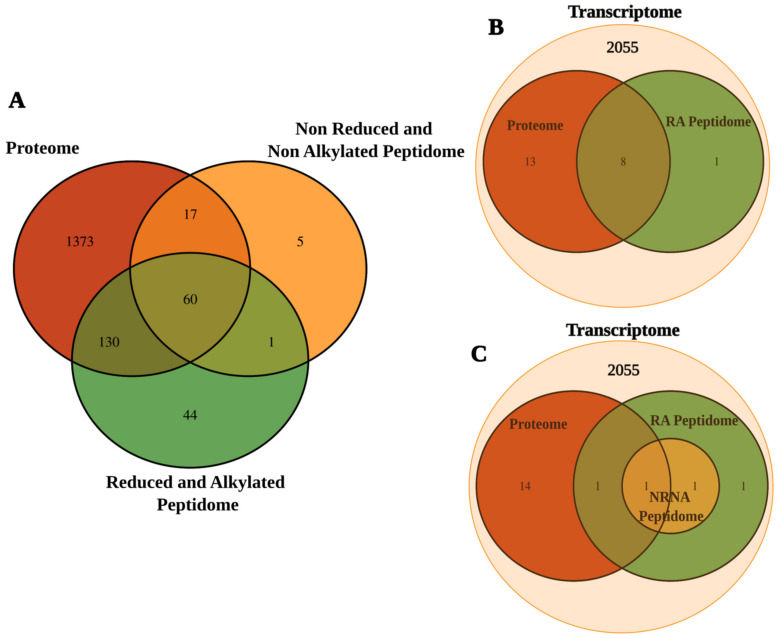
Integrated omics. Comparative analysis of transcripts confirmed by proteomics and peptidomics (**A**). Number of transcripts to the *Bunodosoma caissarum’s* new toxins identified through similarity with other described toxins in transcriptome and confirmed in proteome peptidome (**B**). Number of transcripts to the putative toxins identified through similarity with other described toxins in transcriptome and confirmed in proteome or peptidome (**C**).

**Figure 7 marinedrugs-21-00197-f007:**
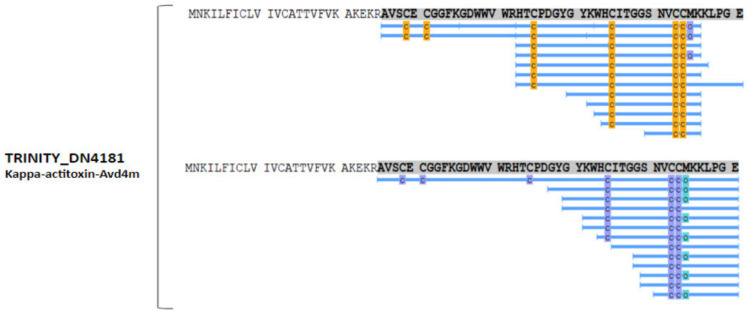
Peptide fragments identified in the proteomics and peptidomics experiments for Kappa-actitocin-Avd4m-like protein. In the sequences, differences in peptides with cysteine residues detected by proteomics (cysteines highlighted in yellow) compared with peptidomics (cysteines highlighted in purple) are observed. The letter *o*, highlighted in green, indicates methionine oxidation.

**Figure 8 marinedrugs-21-00197-f008:**
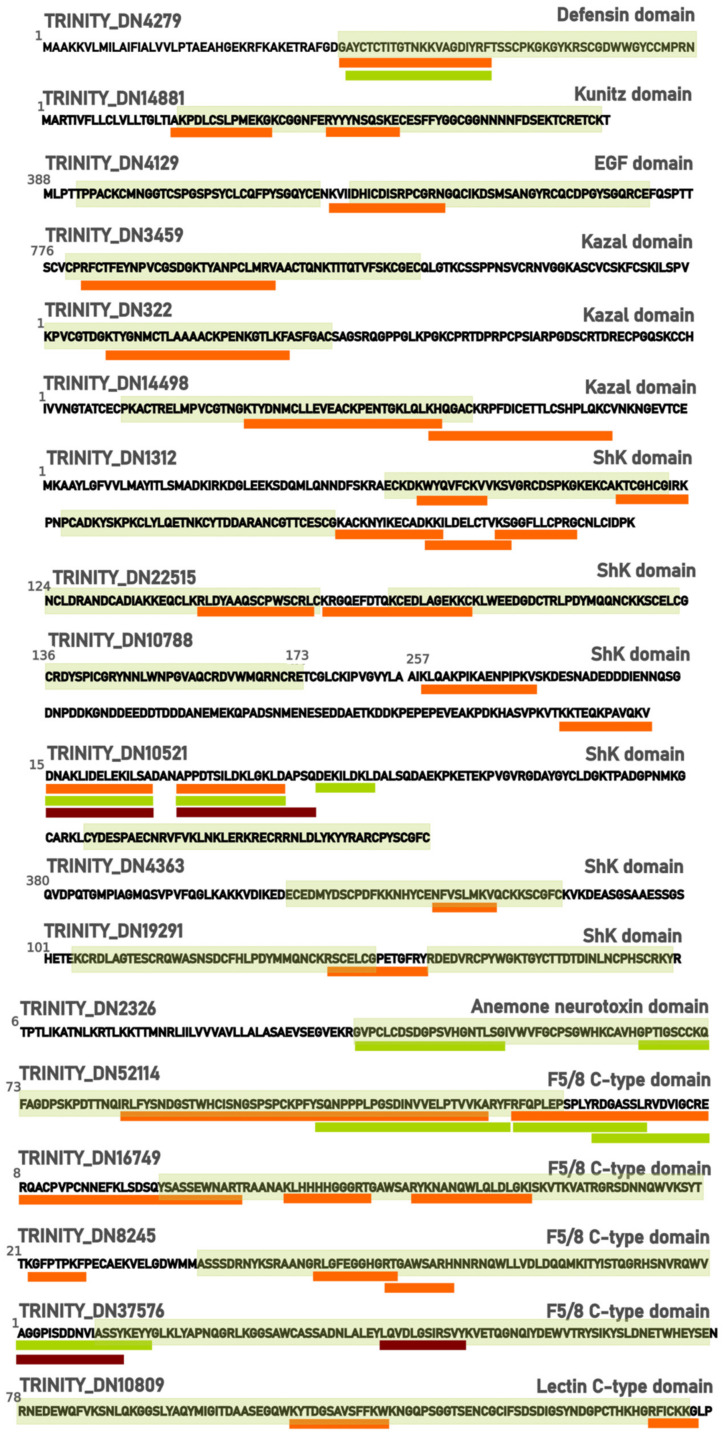
New putative toxins found according to PFAM search of related toxin domains. The transcript sequence is tagged with a light green box representing the corresponding domain. Sequences found with the corresponding transcript in proteome, reduced and alkylated peptidome, and non-reduced and non-alkylated peptidome are represented by orange, green, and dark red boxes, respectively, below the transcript sequence. The alignment of putative toxin transcripts with the three best matches in NCBI with BLASTp is shown in Appendix A.

**Figure 9 marinedrugs-21-00197-f009:**
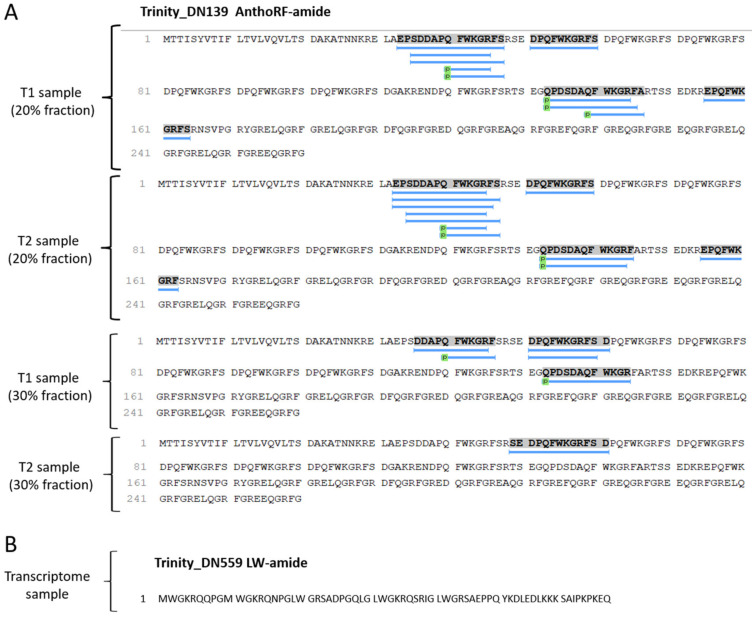
Neuropeptides identified in the tentacles of *B. caissarum*. In (**A**) peptides identified in the Antho-RFamide protein sequence. In (**B**) LWamide protein sequence.

**Table 1 marinedrugs-21-00197-t001:** Trinity and TransDecoder metrics.

	Trinity	TransDecoder
Transcripts	186,978	150,493
N50	1900	501
Genes	111,386	61,465

**Table 2 marinedrugs-21-00197-t002:** Most detected peptides of mucus samples in reduced and alkylated peptidome.

TRINITY	Name	Peptide	Mucus Sample Area	Tentacle Sample Area
TRINITY_DN1213	-	DCRGKHCQTGPFGD	9.06E+09	4.84E+08
TRINITY_DN58641	-	TLASSIQCVGKCKIKTSSGQCRTDLRCMLANKGAS	1.70E+09	3.88E+08
TRINITY_DN14686	U-actitoxin-Bcs2a	GLPCDCHGHTGTYWLNYYSKCPKGYGYTGRCRYLVGSCCYK	1.32E+09	-
TRINITY_DN3092	-	MATSCRKCKPGYGCWAVPCPKR	1.09E+09	-
TRINITY_DN557	Lamin-A	EELSFKRSVYDKE	5.92E+08	-
TRINITY_DN1213	-	DCRGKHCQTGPF	3.56E+08	1.38E+06
TRINITY_DN14686	U-actitoxin-Bcs2a	GLPCDCHGHTGTY	2.59E+08	-
TRINITY_DN3092	-	MATSCRKCKPGYGCWAVPCPKR	2.16E+08	-
TRINITY_DN3092	-	ATSCRKCKPGYGCWAVPCPKR	1.71E+08	-
TRINITY_DN14686	U-actitoxin-Bcs2a	WLNYYSKCPKGYGYTGRCRYLVGSCCYK	1.68E+08	-

**Table 3 marinedrugs-21-00197-t003:** Most detected peptides of tentacle samples in reduced and alkylated peptidome.

TRINITY	Name	Peptide	Mucus Sample Area	Tentacle Sample Area
TRINITY_DN1213	-	DCRGKHCQTGPFGD	9.06E+09	4.84E+08
TRINITY_DN282	Histone H2B	LPGELAKHAVSEGTKAVTKYTSSK	-	3.99E+08
TRINITY_DN58641	-	TLASSIQCVGKCKIKTSSGQCRTDLRCMLANKGAS	1.7E+09	3.88E+08
TRINITY_DN3037	-	NPPYEEILEPAFFHIR	-	2.61E+08
TRINITY_DN10521	-	APPDTSILDKLGKL	7.26E+06	2.13E+08
TRINITY_DN12562	-	DEAGLLKYKTAAGAALVNERLKNLAERY	-	1.81E+08
TRINITY_DN3037	-	QPPFLGGPAYFHIR	-	1.52E+08
TRINITY_DN282	Histone H2B	LLLPGELAKHAVSEGTKAVTKYTSSK	8.66E+06	1.52E+08
TRINITY_DN25490	U-actitoxin-Aeq6a	KERCDLLGDPCVKG	-	1.51E+08
TRINITY_DN1828	-	AAAYVCDVARNLDCSAH	-	1.43E+08

## Data Availability

The data are contained within the article, and mass spectrometry supporting information (raw files) can be downloaded at: http://massive.ucsd.edu/ProteoSAFe/status.jsp?task=1e0eecc81e4d4d3fae4e6260b15c839d.

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
