# Peer review of "Multiomic Approach for Bioprospection: Investigation of Toxins and Peptides of Brazilian Sea Anemone Bunodosoma caissarum"

_marinedrugs, 2023, doi:10.3390/md21030197_

Round 1
Reviewer 1 Report
The manuscript "Multiomics approach for bioprospection: Search of new toxins of Brazilian sea anemone Bunodosoma caissarum" by Mazzi Esquinca et al. describes the transcriptome, proteome, and peptidome of tentacles and mucus of the anemone, with particular emphasis on the proteome and peptidome. The research contributes to understanding the diversity of sea anemone venom components and the composition of other polypeptides found in anemone tentacles and mucus. However, I have some specific points of this investigation that should be addressed.
Title: please re-write the title; the results mostly report polypeptides that have already been reported in other omics studies of other sea anemone species and, to a lesser extent, new polypeptides.
Line 55: replace phylum cnidarians with Phylum Cnidaria
In figure 1: the nomenclature of the subcategories indicated in the graph does not correspond to the nomenclature in the figure caption. Delete the word "Legend" in the graph. Place the line of the axes on the graph.
In the text Cnidaria toxin, Cnidaria species is mentioned, the word Cnidaria is in lower case, and the term Cnidaria phylum, the correct term is Phylum Cnidaria.
Please edit the tables to make them more readable.
Figure 2. The description of the proteins should be included in the text, not in the caption. incorporate references
Figure 3, in the graphs, incorporate the axis of the independent variables and appends the name of the axis of the dependent variables.
Graphic styles should be standardized; eliminate the word "Legend" from the graphics.
Section 2.5, it is not necessary to append the universal nomenclature of amino acids in parentheses.
Figure 6. The title of the figure is missing
Improve resolution of the figures where it presents alignments
The discussion lacks incorporating references to support the information
The discussion lacks incorporating references to support the information.
Section 3.1 incorporate it in the results since this section is, for the most part, descriptive
In supplementary material, the first part indicates Legend; is it a table? If it is a table, indicate the table number and title.
In Supplementary Figure 1: Did the consistency of the sample with mucus interfere with the passage of polypeptides of less than 10 kDa through the membrane? How did you check the complexity of the sample before analyzing it by MS? Did the sample that was retained on the filter contain peptides of less than 10 kDa?
How do you validate this step for efficient separation of samples for peptidomics and proteomics? In this figure, the result of this protocol of the toxins obtained, not all of them is new toxins.
Section 2.3 refers to the mucus and tentacle peptidome. However, most of these peptides are part of proteins. Strictly speaking, the peptidome refers to endogenous peptides, as opposed to peptides present in a sample as a product of proteolysis.
Section 3.1 Some proteins are described in well-defined sections; the manuscript must have homogeneity in its structure.
In the proteomics and peptidomics analysis, the coverage percentage is missing.
Author Response
Thank you for your very nice and critical comments, that were all very appreciated. Below, please, find our attempt to answer your main questions.
Title: please re-write the title; the results mostly report polypeptides that have already been reported in other omics studies of other sea anemone species and, to a lesser extent, new polypeptides.
Answer: Thanks, The title has been changed.
- Line 55: replace phylum cnidarians with Phylum Cnidaria
Answer:Thanks. The term has been corrected.
- In figure 1: the nomenclature of the subcategories indicated in the graph does not correspond to the nomenclature in the figure caption. Delete the word "Legend" in the graph. Place the line of the axes on the graph. In the text Cnidaria toxin, Cnidaria species is mentioned, the word Cnidaria is in lower case, and the term Cnidaria phylum, the correct term is Phylum Cnidaria.
Answer: We did all the requested corrections regarding the figure, text and caption.
- Please edit the tables to make them more readable.
Answer: We used the templates/layouts recommended by Marine Drugs to assemble the tables.
- Figure 2. The description of the proteins should be included in the text, not in the caption. incorporate references
Answer: Thanks. The description of proteins was moved from caption to text and references added.
- Figure 3, in the graphs, incorporates the axis of the independent variables and appends the name of the axis of the dependent variables. Graphic styles should be standardized; eliminate the word "Legend" from the graphics.
Answer: All suggestions regarding Figure 3 were made.
- Section 2.5, it is not necessary to append the universal nomenclature of amino acids in parentheses. Figure 6. The title of the figure is missing. Improve resolution of the figures where it presents alignments.
Answer: Thanks. The quality of the figures have been improved in this new version of the manuscript.
8)The discussion lacks incorporating references to support the information Section 3.2.1 incorporate it in the results since this section is, for the most part, descriptive
Answer: The section was checked and new references were incorporated.
9) In supplementary material, the first part indicates Legend; is it a table? If it is a table, indicate the table number and title.
Answer: Thanks. I am sorry for the mistake. The corrected term is “Abbreviations”.
10) In Supplementary Figure 1: Did the consistency of the sample with mucus interfere with the passage of polypeptides of less than 10 kDa through the membrane? How did you check the complexity of the sample before analyzing it by MS? Did the sample that was retained on the filter contain peptides of less than 10 kDa?How do you validate this step for efficient separation of samples for peptidomics and proteomics? In this figure, the result of this protocol of the toxins obtained, not all of them are new toxins.
Answer: Probably the consistency of mucus interferes with the passage of some toxins and other polypeptides, because toxins were also found in the fraction above 10 KDa. However, it was possible to identify that some toxins and other polypeptides were present exclusively in the peptidome experiments with fractions below 10 KDa. In this study we used what we had available in terms of separation and based on what is currently most used. Every protocol has its limitations, however we believe that the separations used helped for a more efficient identification of sequences and they are complementary.
11) Section 2.3 refers to the mucus and tentacle peptidome. However, most of these peptides are part of proteins. Strictly speaking, the peptidome refers to endogenous peptides, as opposed to peptides present in a sample as a product of proteolysis.
The peptidome experiments of this section were performed without digestion with trypsin and with a rapid inactivation protocol to avoid proteolysis during extraction. These fragments are generated during protein metabolism, mainly generated by the proteolytic proteasome complex.In recent decades several research groups have demonstrated biological activity of these peptide fragments.
12) Section 3.1 Some proteins are described in well-defined sections; the manuscript must have homogeneity in its structure.
Answer: The discussion sections have been improved.
13) In the proteomics and peptidomics analysis, the coverage percentage is missing.
Answer: The coverage percentage was added in the Supplementary table 2.
Reviewer 2 Report
In the manuscript entitled "Multiomics approach for bioprospection: Search of new toxins of Brazilian sea anemone Bunodosoma caissarum”, the authors have characterized the protein composition of the tentacles and mucus of Bunodosoma caissarum using a multiomics approach. The authors need to address the following points:
Major comments
1. The paper would be much more interesting if a more formal comparison was presented between the transcriptome results and peptidome/proteome. Either a table or graphical illustration comparing the relative abundance of the toxin transcripts vs the peptidome/proteome.
2. Authors indicate the percent identity of B. caissarum’s new toxins with other described toxins in Tables 4 and 5. It is unclear which organism the authors used to compare the similarity with, only the sea anemone Actinia tenebrosa or other related species? Please clarify.
3. The authors should provide the multiple alignments of new putative toxins neither in figures nor as supplementary material.
Minor criticisms
4. Authors should a comma to separate thousands (e.g. 1900 in Table 1; line 143 – (66,2%); line 144 – (34,5%)).
5. The caption of Figure 1. “Percentage of annotated genes in subcategories according…”. This caption is not supported by the value in the graph. It seems that the numbers on the graph are the numbers of genes, not percentages. The labeled color legends (FM and PB) are not related to the caption. Also, the authors should spell out “WEGO” for clarity.
6. The authors should use percentages of toxins found in the transcriptomic or proteomic data in the pie charts of Figures 2A and 3A-B, respectively.
7. Figures 4 and 6 could be significantly improved. It would be critical for understanding the results. The authors should clarify what the values in Figure 4 are. What is highlighted in green in Figure 6? Please clarify.
8. Authors should write the genus in full when it is only first used (e.g., Line 81: Actinia tenebrosa, line 112 should be A. tenebrosa).
9. Authors should spell out “PFAM” for the first used.
10. Figure 7: Authors should include amino acid residues for all toxins, as authors indicate for the TRINITY_DN10788.
Author Response
Thank you for your very nice and critical comments, that were all very appreciated. Below, please, find our attempt to answer your main questions.
- The paper would be much more interesting if a more formal comparison was presented between the transcriptome results and peptidome/proteome. Either a table or graphical illustration comparing the relative abundance of the toxin transcripts vs the peptidome/proteome.
Answer: A graphical illustration was added to the discussion.
- Authors indicate the percent identity of B. caissarum’s new toxins with other described toxins in Tables 4 and 5. It is unclear which organism the authors used to compare the similarity with, only the sea anemone Actinia tenebrosa or other related species? Please clarify.
Answer: Part of the toxin codes indicates the species according to the nomenclature. We indicate the acronyms and the name of the species at the end of the table.
- The authors should provide the multiple alignments of new putative toxins neither in figures nor as supplementary material.
Answer: Multiple alignments are shown in the Supplementary Figure 6.
Minor criticisms
- Authors should use a comma to separate thousands (e.g. 1900 in Table 1; line 143 – (66,2%); line 144 – (34,5%)).
Answer: Thanks. We did the requested corrections.
- The caption of Figure 1. “Percentage of annotated genes in subcategories according…”. This caption is not supported by the value in the graph. It seems that the numbers on the graph are the numbers of genes, not percentages. The labeled color legends (FM and PB) are not related to the caption. Also, the authors should spell out “WEGO” for clarity.
Answer: We did all the requested corrections.
- The authors should use percentages of toxins found in the transcriptomic or proteomic data in the pie charts of Figures 2A and 3A-B, respectively.
Answer:The figure indicated by the reviewer was modified.
- Figures 4 and 6 could be significantly improved. It would be critical for understanding the results. The authors should clarify what the values in Figure 4 are. What is highlighted in green in Figure 6? Please clarify.
Answer: The quantification values in Figure 4 were normalized using programma R. This figure is a heat map
The letter highlighted in green in figure 6, demonstrates the presence of methionine oxidation modification. Figure 6 changed to Figure 7.
- Authors should write the genus in full when it is only first used (e.g., Line 81: Actinia tenebrosa, line 112 should be A. tenebrosa). Authors should spell out “PFAM” for the first used.
Answer: We did all the requested corrections. The full name of the abbreviation was indicated in the text.
- Figure 7: Authors should include amino acid residues for all toxins, as authors indicate for the TRINITY_DN10788.
Answer: Amino acid residues were included for the other transcripts.
Reviewer 3 Report
The manuscript is an excellent extensive study, performed at a high modern scientific and experimental level. I wish the authors further success in the study of unique marine organisms - producers of biologically active peptides and proteins, in order to create a basis for obtaining new pharmacological agents and biochemical tools
Author Response
Thank you for the comments.